# The Assessment of Anti-Melanoma Potential of Tigecycline—Cellular and Molecular Studies of Cell Proliferation, Apoptosis and Autophagy on Amelanotic and Melanotic Melanoma Cells

**DOI:** 10.3390/cells12121564

**Published:** 2023-06-06

**Authors:** Jakub Rok, Justyna Kowalska, Zuzanna Rzepka, Dominika Stencel, Anna Skorek, Klaudia Banach, Dorota Wrześniok

**Affiliations:** Department of Pharmaceutical Chemistry, Faculty of Pharmaceutical Sciences in Sosnowiec, Medical University of Silesia in Katowice, 4 Jagiellońska Str., 41-200 Sosnowiec, Poland; jkowalska@sum.edu.pl (J.K.); zrzepka@sum.edu.pl (Z.R.); kbanach@sum.edu.pl (K.B.); dwrzesniok@sum.edu.pl (D.W.)

**Keywords:** tigecycline, melanoma, apoptosis, autophagy, cell cycle, cell proliferation

## Abstract

High mortality, aggressiveness, and the relatively low effectiveness of therapy make melanoma the most dangerous of skin cancers. Previously published studies presented the promising therapeutic potential of minocycline, doxycycline, and chlortetracycline on melanoma cells. This study aimed to assess the cytotoxicity of tigecycline, a third-generation tetracycline, on melanotic (COLO 829) and amelanotic (A375) melanoma cell lines. The obtained results showed that tigecycline, proportionally to the concentration and incubation time, efficiently inhibited proliferation of both types of melanoma cells. The effect was accompanied by the dysregulation of the cell cycle, the depolarization of the mitochondrial membrane, and a decrease in the reduced thiols and the levels of MITF and p44/42 MAPK. However, the ability to induce apoptosis was only found in COLO 829 melanoma cells. A375 cells appeared to be more resistant to the treatment with tigecycline. The drug did not induce apoptosis but caused an increase in LC3A/B protein levels—an autophagy marker. The observed differences in drug action on the tested cell lines also involved an increase in p21 and p16 protein levels in melanotic melanoma, which was related to cell cycle arrest in the G1/G0 phase. The greater sensitivity of melanotic melanoma cells to the action of tigecycline suggests the possibility of considering the use of the drug in targeted therapy.

## 1. Introduction

Melanoma is thought to be the most serious and deadly type of skin cancer. This cancer originates from highly differentiated melanin-producing cells called melanocytes. The environmental risk factors for melanoma development include exposure to ultraviolet (UV) radiation, heavy metals, or pesticides. Additionally, individual factors, including innate genetic ones, play a significant role. Fair skin, blond or red hair, and light-colored eyes are examples of such factors [1,2,3]. The formation of melanoma also results in acquired genetic and epigenetic changes. The BRAF V600E mutation in the MAPK cascade is one of the most characteristic changes in melanoma and is observed at an early stage of the disease [4].

Despite significant knowledge about the causes of and risk factors for the development of melanoma, it remains a fairly common cancer, ranking as the 19th-most-frequently diagnosed cancer worldwide. In addition, melanoma is associated with high mortality rates, accounting for 65% of skin-cancer-related deaths, while representing less than 5% of diagnosed skin cancers [5]. The incidence rates of melanoma continue to rise each year, particularly among young people [6]. Early detection of melanoma is crucial, with diagnosis based on the morphological and histopathological assessment of the lesion. Morphological assessment relies on Breslow’s classification system, which considers the size of the pathological lesion, while histopathological assessment allows us to distinguish between types of melanoma, including superficial spreading melanoma (SSM), lentigo maligna melanoma (LMM), and nodular melanoma (NM) [7]. Treatment for melanoma typically involves surgery, with radiation and pharmacotherapy used in cases where surgery is insufficient. Pharmacotherapeutic methods include chemotherapy (such as dacarbazine, temozolomide, or fotemustine), targeted therapy (such as vemurafenib or dabrafenib), and checkpoint immunotherapy (such as nivolumab, pembrolizumab, or ipilimumab) [2,8,9,10]. Although the last two therapeutic strategies have significantly prolonged the lives of patients, completely curing melanoma is still a considerable challenge. Therefore, new, more effective, and safer drugs are being sought, which would form the basis or an element of a complex therapy aimed at improving the quality of life and achieving complete recovery.

According to previous scientific reports, tetracycline antibiotics, such as doxycycline and minocycline, showed promising therapeutic effects in cancer treatment, including melanoma [11,12]. The molecular mechanisms of their anticancer action involve, among others, the inhibition of matrix metalloproteinase, HSP90, as well as the ERK/MAPK and PAR1/FAK/PI3K/AKT pathways. They also increase the levels of the proapoptotic protein BAX and decrease the levels of the anti-apoptotic protein BCL-2. These antibiotics also reduce proliferation markers such as Ki67 and PCNA, as well as the tumor necrosis factor TNF-α, by inhibiting the activation of NF-κB [13,14,15,16,17,18]. As a result, these drugs can arrest cell proliferation, induce apoptosis, and inhibit cell migration and the development of cancer stem cells. Furthermore, it was demonstrated that doxycycline and minocycline inhibit angiogenesis by affecting IL-8 and the translational suppression of HIF-1α, respectively [19,20]. 

Tigecycline, a third-generation tetracycline that includes a tert-butylglycylamide substituent at the C-9 position, was found to have both antibiotic and non-antibiotic properties and, among other things, anticancer activity [21]. The drug was approved for use in medicine in 2005 for complicated and resistant bacterial infections due to its extended-spectrum antibiotic properties [22,23]. Its bacteriostatic activity results from the ability to inhibit bacterial protein synthesis by binding to the 30S subunit of the bacterial ribosome [24]. Moreover, there are reports indicating that tigecycline may have potential in the treatment of various types of cancer, including ovarian, breast, cervical, thyroid, lung, pancreatic, stomach, and skin cancers [16,25]. The main anticancer effects of tigecycline are similar to those observed for doxycycline and minocycline. These effects involve the inhibition of cell proliferation, induction of apoptosis, influence on autophagy activation, blockade of angiogenesis, and suppression of migration. The molecular mechanisms underlying these actions include the significant inhibition of the mitochondrial translation and signaling pathways related to p21^CIP1/Waf1^, Wnt/β, Akt, and mTOR kinase [26,27,28]. A previously conducted study on melanoma cells demonstrated that tigecycline prevented epithelial–mesenchymal transition and arrested the cell cycle at the G0/G1 phase. These actions were dependent on the cyclin-dependent kinase inhibitor p21^CIP1/Waf1^ [29].

Based on previous studies demonstrating the anti-melanoma activity of some tetracyclines, an attempt was made to evaluate the activity of tigecycline against melanotic and amelanotic melanoma cells. 

## 2. Materials and Methods

### 2.1. Chemicals and Reagents

Tygacil (50 mg tigecycline C_29_H_39_N_5_O_8_ × 10 vials) was obtained from Pfizer (New York, NY, USA). Phosphate-buffered saline solution (PBS), Tween-20, RIPA Buffer, PVDF membranes, SIGMAFAST™ Protease Inhibitor Cocktail Tablet, Phosphatase Inhibitor Cocktail 3, Fibroblast Growth Medium, Triton X-100 solution, paraformaldehyde, Phalloidin-Atto 565, Bovine Serum Albumin (BSA), Tris-Buffered Saline (TBS), and the antibiotics penicillin and amphotericin B were acquired from Sigma-Aldrich Inc. (Taufkirchen, Germany). Roswell Park Memorial Institute (RPMI) 1640 medium, M-254 medium, and Human Melanocyte Growth Supplement-2 (HMGS-2) were obtained from Cascade Biologics (Portland, OR, USA). Dulbecco’s Modified Eagle Medium (DMEM), ECL Western Blotting Substrate, Hoechst 33342, Pierce™ BCA Protein Assay Kit, secondary antibody—Alexa Fluor 488, and SYTO Deep Red Nucleic Acid Stain were acquired from Thermo Fisher Scientific (Waltham, MA, USA). Neomycin was obtained from Amara (Kraków, Poland). Trypsin/EDTA and Fetal Bovine Serum (FBS) were obtained from Cytogen (Zgierz, Poland). Cell Proliferation Reagent WST-1 was purchased from Roche GmbH (Mannheim, Germany). FITC-labeled annexin V was purchased from Biotium (Fremont, CA, USA). Via-1-Cassettes™ (contain acridine orange and DAPI fluorophores), NC-Slides™ A2 and A8, Annexin V Binding Buffer, Apoptosis Wash Buffer and the staining reagents Solution 3 (1 µg/mL DAPI, 0.1% triton X-100 in PBS), Solution 5 (400 µg/mL VitaBright48™, 500 µg/mL propidium iodide, 1.2 µg/mL acridine orange in DMSO), Solution 7 (200 µg/mL JC-1), Solution 8 (1 µg/mL DAPI in PBS), Solution 15 (500 µg/mL Hoechst 33342), and Solution 16 (500 µg/mL propidium iodide) were obtained from ChemoMetec (Lillerød, Denmark). Primary rabbit monoclonal antibodies, anti-GAPDH, anti-LC3A/B, anti-MITF, anti-p16, anti-21, and anti-p44/p42 were obtained from Cell Signaling (Danvers, MA, USA). Caspase 3/7 Assay Kit was purchased from ImmunoChemistry Technologies (Bloomington, MN, USA). The remaining chemicals were acquired from POCH S.A. (Gliwice, Poland) or Sigma-Aldrich (Taufkirchen, Germany).

### 2.2. Cell Culture

The human melanoma cell lines COLO 829 (cutaneous melanoma cells with fibroblast morphology, having mutations of the BRAF, CDKN2A, PTEN, and TERT genes) and A375 (skin amelanotic melanoma cells with epithelial morphology, having mutations of the NRAS, BRAF, CDKN2A, and TERT genes) were used in the conducted studies. Both cell lines were obtained from ATCC (CRL-1974TM, Manassas, VA, USA). Cells were cultured in a suitable culture medium supplemented with 10% fetal bovine serum, at 37 °C in 5% CO_2_. Melanotic melanoma cells COLO 829 were cultivated in RPMI medium supplemented with L-glutamine, and amelanotic melanoma cells A375 were cultured in DMEM medium. Human dermal, neonatal, fibroblasts (HDF) (Sigma Aldrich Inc., St. Louis, MO, USA) and human epidermal melanocytes, neonatal, darkly pigmented (HEMn-DP) (Cascade Biologics, Portland, OR, USA) were cultured in Fibroblast Growth Medium and M-254 medium with HMGS-2, respectively. Penicillin (10,000 U/mL), neomycin (10 μg/mL), and amphotericin B (0.25 mg/mL) were prophylactically added to the culture media.

### 2.3. Screening Cytotoxicity Assessment

The proliferation of cells treated with tigecycline was determined using a test Cell Proliferation Reagent I (WST-1). Melanoma cells were seeded in 96-well plates at a density of 2.5 × 10^3^ cells/well and incubated in a culture medium for 24 h. In turn, fibroblasts and melanocytes were seeded at a density 5 × 10^3^ cells/well and cultured for 48 h before the treatment. Next, the medium was removed, and solutions of tigecycline in culture medium were added. The reagent WST-1 was added to each well (10 µL/well) 3 h before the measurement. The absorbance at wavelengths of 440 nm and 650 nm was measured using a microplate reader Infinite 200 PRO (TECAN, Männedorf, Switzerland). The mean absorbance difference for the controls was normalized as 100%, while the results for the samples were expressed as a percentage of the control.

### 2.4. Analysis of Cell Number and Cell Viability

The assessment of tested cell population considering the viability and number of cells was made using the NucleoCounter^®^ NC-3000™ fluorescence image cytometer (ChemoMetec, Lillerød, Denmark). The cells, after detachment by trypsinization, were centrifuged and suspended in the recommended culture medium. Next, the sample of cell suspension was loaded into the Via1-Cassette™ and stained with fluorescent dyes contained in the cassette: DAPI and acridine orange. The immediate analysis allowed to determine the percentage of dead cells (DAPI only penetrates through a damaged cell membrane) as well as total cell population (acridine orange penetrates into all cells). The analysis was performed according to the “Cell Viability and Cell Count Assays” protocol.

### 2.5. Cell Cycle Analysis

Cell cycle analysis was performed using a NucleoCounter^®^ NC-3000™ fluorescent imaging cytometer. The assay is based on the quantitative measurement of DNA in cells that have been previously fixed and stained with DAPI reagent. After 48-h incubation with tigecycline, cells were harvested by trypsinization and centrifuged. Then, 1 × 10^6^ cells were suspended in 0.5 mL PBS and fixed by adding 4.5 mL of ice-cold 70% ethanol. The prepared samples were kept for 24 h at 2–8 °C. Subsequently, cells were centrifuged, washed with PBS, and suspended in 0.5 mL of Solution 3. Following 5 min of incubation at 37 °C, analysis was performed according to the protocol “Fixed Cell Cycle—DAPI Assay”. The obtained results were analyzed in accordance with the recommendations of the manufacturer.

### 2.6. Reduced Thiols Analysis

The assay is based on determining the level of reduced intracellular thiols using the VitaBright-48™ dye, which reacts with thiols to generate a fluorescent product. Melanoma cells were treated with tigecycline for 24 and 48 h, trypsinized, counted, and suspended in PBS. The cells were then stained using Solution 5, loaded into 8-chamber NC-Slides A8™, and measured using the “Vitality Assay” protocol on a fluorescent imaging cytometer NucleoCounter^®^ NC-3000™.

### 2.7. Mitochondrial Membrane Potential Assay

The potential of the mitochondrial membrane in melanoma cells was assessed using the NucleoCounter^®^ NC-3000™ fluorescent imaging cytometer (ChemoMetec, Lillerød, Denmark). Cells were treated with tigecycline for 24 h and 48 h. After treatment, cells were harvested and counted. Then, 1 × 10^6^ cells were stained with Solution 7, which contained JC-1. In healthy cells (with high mitochondrial membrane potential), the JC-1 dye is localized in the mitochondria and forms aggregates emitting red fluorescence. In apoptotic cells (with low mitochondrial membrane potential), the dye is released into the cytoplasm in the form of monomers, emitting green fluorescence. Following a 10-min incubation at 37 °C, cells were centrifuged and washed twice with PBS. The obtained cell precipitates were suspended in 0.25 mL of Solution 8 and analyzed according to the “Mitochondrial Potential Assay” protocol.

### 2.8. Annexin V Assay

The Annexin V assay is a quantitative test that determines the number of cells undergoing apoptosis. Annexin V protein binds to apoptotic cells via phosphatidylserine, which is translocated from the inner side to the outer side of the cell membrane during apoptosis. After a 48-h treatment with tigecycline, cells were trypsinized and counted. Then, 3 × 10^5^ cells were stained with FITC-labeled Annexin V and Solution 15 for 15 min at 37 °C. Subsequently, cells were centrifuged, washed twice with Annexin V Binding Buffer, and suspended in 100 μL of the buffer supplemented with 10 μg/mL propidium iodide (Solution 16). The obtained cell precipitates were analyzed according to the “Annexin V Assay” protocol using a fluorescent imaging cytometer.

### 2.9. Caspase 3/7 Activity Assay

The activity of caspase 3/7 was cytometrically analyzed. The assay is based on fluorochrome-labeled inhibitor of caspase (FLICA): Caspase 3/7 Assay Kit. Melanoma cells, after the treatment, were harvested, counted, and suspended in PBS (4 × 10^6^ cells/mL). In the next step, 5 μL of FLICA reagent and 2 μL of Hoechst 33342 were added to 93 µL of cell suspensions and then incubated for 60 min at 37 °C. Afterwards, the samples were washed twice with 400 µL of Apoptosis Wash Buffer, centrifuged, and resuspended in 100 µL of the buffer supplemented with 10 μg/mL propidium iodide. The analysis was performed using the NucleoCounter^®^ NC-3000™ according to the “Caspase Assay” protocol.

### 2.10. Western Blotting Analysis

Melanoma cells were exposed to tigecycline for 48 h and then lysed using RIPA buffer containing phosphatase and protease inhibitors. The total protein concentration in all lysates was determined using the Pierce™ BCA Protein Assay Kit, in accordance with the instructions of the manufacturer. Protein extracts (40 µg/lane) were separated by SDS-PAGE and transferred onto a PVDF membrane. The membrane was blocked for 1 h in a blocking buffer (5% non-fat milk in TBST), washed with TBST (Tris-Buffered Saline with Tween 20), and incubated overnight at 4 °C with primary antibodies against LC3A/B (1:1000), MITF (1:500), p44/42 (1:1000), or GAPDH (1:1000). Subsequently, the membranes were washed and incubated with a horseradish peroxidase-conjugated anti-rabbit secondary antibody for 1.5 h at room temperature. The protein signals were detected using an ECL chemiluminescence reagent. The analysis was performed using a G:Box Chemi-XT4 Imaging System and GeneTools Software (Syngene, Cambridge, UK). GAPDH was used to normalize for loading variations.

### 2.11. Confocal Microscopy Imaging

Imaging of melanoma cells was performed using the laser confocal microscope Nikon Eclipse Ti-E A1R-Si and Nikon NIS Elements AR software. A375 and COLO 829 cells were cultured in sterile cover slips in Petri dishes. After 48-h exposure to tigecycline, the cells were fixed with 4% paraformaldehyde and permeabilized with 0.1% Triton X-100. Afterwards, fixed cells were treated with glycine and BSA solutions and then were incubated with primary anti-p16 (1:800) and anti-p21 (1:400) rabbit antibodies overnight at 4 °C. In the next step, the samples were stained with Alexa Fluor 488 conjugated with the secondary antibody (1:200), SYTO Deep Red Nucleic Acid Stain (1:100), and Phalloidin–Atto 565 (0.6 µM). The reagents allowed for imaging of p16 and p21 proteins, as well as nucleus and actin cytoskeleton, respectively. Finally, fixed and stained cells on cover slips were mounted onto microscopic glass slides.

### 2.12. Statistical Analysis

The obtained results were subjected to statistical analysis using GraphPad Prism 6.01, which included calculating arithmetic means and standard deviations (SD), checking the consistency of result distribution with normal distribution (Kolmogorov–Smirnov test), and determining if the differences between the compared groups were statistically significant. To achieve this, the assumption of homogeneity of variance (Brown–Forsythe test) was verified. Additionally, the results were analyzed by one-way and two-way ANOVA, as well as Dunnett’s and Tukey’s tests. In all cases, statistical significance was determined by a *p*-value less than 0.05.

## 3. Results

### 3.1. Screening Assessment of Cytotoxic Effect of Tigecycline on Melanoma Cells

The initial assessment of the cytotoxic effect of tigecycline on melanoma cells was made using a WST-1 assay. This colorimetric test allows for the determination of the changes in the number of metabolically active cells. Experiments were conducted on amelanotic (A375) and melanotic (COLO 829) melanoma cell lines. The obtained results (Figure 1) showed that tigecycline proportionally reduced the number of living and metabolically active cells to the drug concentration as well as the time of incubation. Thus, the lowest values were noticed after 72-h treatment with tigecycline in a concentration of 400 µM. In this case, the decrease in the metabolically active cells was 75.2% for the A375 cell line and 97.8% for the COLO 829 cell line. It was concluded that melanotic melanoma appeared to be more sensitive to tigecycline than amelanotic melanoma. The calculated EC_50_ values for the COLO 829 cells were 91.5 µM, 21.0 µM, and 19.2 µM, respectively, for treatment times of 24 h, 48 h, and 72 h. In turn, the corresponding results for the A375 cell line were 237.3 µM, 82.0 µM, and 39.1 µM.

To assess control normal cell lines, preliminary analyses were conducted on human dermal fibroblasts and darkly pigmented human epidermal melanocytes. The test was performed after 48 h of incubation with tigecycline in concentrations ranging from 1 to 400 µM. The obtained results (Figure 2) indicated that the drug, similar to melanoma cells, proportionally reduced the number of metabolically active cells with increasing concentration. However, the calculated EC_50_ values for the 48-h treatment were higher compared to those of the melanotic and amelanotic melanoma cells: 126.2 µM for HDF and 148.0 µM for HEMn-DP. It is also worth noting that only in normal darkly pigmented melanocytes did tigecycline at concentrations up to 25 µM not exhibit cytotoxic effects.

It is worth noting that the results of the WST-1 test are an outcome of both the antiproliferative and cytotoxic effects, leading to cell death. Therefore, additional analyses more precisely indicating the mechanism of action of the drug are necessary. Considering the results, the next experiments were conducted using selected drug concentrations: 200 µM and 400 µM for the A375 cell line and 100 µM and 200 µM for the COLO 829 cell line.

### 3.2. Influence of Tigecycline on Proliferation, Viability, and Apoptosis of Melanoma Cells

The next stage of this study aimed to assess the mechanism of the cytotoxic effect of tigecycline in more detail. For this purpose, the cell number, viability, and apoptosis were analyzed. The obtained results are presented in Figure 3 and Figure 4. It was found that the treatment with tigecycline caused the effective inhibition of cell division. The action was observed in cultures of amelanotic and melanotic melanoma cells. The estimation of the cell number showed that cells in control cultures proliferated and practically doubled the number of days, whereas the number of treated cells did not increase. The cell count coefficient for A375 cells after 24-h and 48-h treatments was 0.81 and 0.75, respectively (Figure 3b). The corresponding results for COLO 829 cells treated with tigecycline were 0.65 and 0.64, respectively (Figure 4b). The significantly reduction in the cell number in treated samples could also be observed in the microscopic images (Figure 3a and Figure 4a). The effect was noticed for both tested cell lines. The treatment of melanoma cultures with tigecycline decreased the number of cells and caused changes in the morphology, especially in the melanotic melanoma samples. It was observed that tigecycline-treated COLO 829 cells exhibited a tendency toward a rounded and spherical morphology, as well as independent growth, without apparent intercellular interactions.

It is worth noting that tigecycline did not cause a significant increase in the percentage of dead cells in the studied populations. An analysis revealed a decrease in cell viability by about only 7% for A375 cells treated with 400 µM of tigecycline for 48 h (Figure 3c). The percentage of dead COLO 829 cells after 48-h incubation with 200 µM of tigecycline was about 11% (Figure 4c).

The ability of tigecycline to induction cell apoptosis was assessed by annexin V and caspase 3/7 analyses. The obtained results indicated that the tested drug, in a concentration of 400 µM, did not cause apoptosis in A375 cells (Figure 3d,e). No statistically significant difference between controls and treated samples was observed for both tests. In turn, an increase in the percentage of apoptotic cells was found in COLO 829 samples treated with 200 µM of tigecycline for 48 h (Figure 4d,e). Annexin V-positive cells accounted for 24% of the melanotic melanoma population incubated with tigecycline. In turn, the percentage of treated COLO 829 cells with active caspase 3/7 was estimated at 33%.

### 3.3. Tigecycline Decreases the Level of Reduced Thiols in Melanoma Cells

Intracellular thiols, such as glutathione, participate in redox homeostasis. As proton donors, they have the ability to reduce other compounds, including reactive oxygen species. Thus, an increase in the percentage of cells with a low level of reduced thiols indicates redox imbalance and oxidative stress. The tested melanoma cells were analyzed after 24-h and 48-h incubation with tigecycline. The assessment of reduced thiols was cytometrically made using the Cell Vitality Assay. The method is based on a dye that fluoresces after the binding to reduced thiols. The results presented in Figure 5 show that TGC decreased the level of reduced thiols in both tested melanoma cell lines. The effect was proportional to the incubation time as well as the drug concentration. The percentage of cells with a low level of GSH was about 33% and 47% for A375 cells after 48-h treatment with 200 µM and 400 µM of TGC, respectively (Figure 5a,b). The changes observed in COLO 829 melanotic melanoma cells were similar. It was found that 100 µM and 200 µM of tigecycline lowered the level of reduced GSH in about 30% and 42% of cells after 48 h, respectively (Figure 5c,d).

### 3.4. Tigecycline Decreases Mitochondrial Membrane Potential (MMP) in Melanoma Cells

An analysis of mitochondrial membrane potential was based on staining cells with JC-1—a day that accumulated in the matrix of mitochondria with high membrane potential. The accumulation leads to form aggregates of JC-1 that show red fluorescence. In turn, the green fluorescence of JC-1, characteristic for monomeric forms, dominates in cells with depolarized mitochondria. The performed analysis indicated that tigecycline depolarized mitochondria in amelanotic and melanotic melanoma cells in a concentration-dependent manner. The percentage of A375 cells with low MMP after 24-h treatment were estimated at 30% and 37% for 200 µM and 400 µM of TGC, respectively. Prolongation of the incubation time to 48 h caused only a slight, few-percentage increase in the number of amelanotic melanoma cells with low MMP (Figure 6a,b). The time of exposure to TGC was more important in the case of the melanotic melanoma cells. A low level of MMP was found in 9% and 28% of COLO 829 cells incubated with 100 µM of TGC for 24 h and 48 h, respectively. The corresponding results for 200 µM of TGC were 19% and 31% (Figure 6c,d).

### 3.5. Tigecycline Disturbs Cell Cycle of Melanoma Cells

Cell cycle of tested melanoma cells were cytometrically analyzed after 48 h incubation with tigecycline. This method is based on the measurement of the fluorescence emitted by DAPI—a dye that selectively binds to DNA. The obtained results are presented in Figure 7 as representative histograms as well as bar graphs. The performed analysis showed that tigecycline disturbed the cycle of melanoma cells; however, the final effect depended on the type of cells. It was found that the incubation of A375 cells with TGC (Figure 7a,b) resulted in the decrease in the percentage of cells in the G0/G1 phase (79% for control vs. 59% for 400 µM of TGC) as well as in the increase in the number of cells in the S (9% for control vs. 17% for 400 µM of TGC) and G2/M phases (10% for control vs. 18% for 400 µM of TGC). In turn, the incubation of COLO 829 cells with tigecycline (Figure 7c,d) caused an increase in the number of cells in the G1/G0 phase (59% for control vs. 80% for 200 µM of TGC). Moreover, a significant decrease in the percentage of melanotic melanoma cells in the S (16% for control vs. 2% for 200 µM of TGC) and G2/M phases (21% for control vs. 14% for 200 µM of TGC) was observed after the treatment.

### 3.6. Tigecycline Changes the Level of Proteins Regulating Proliferation and Survival of Melanoma Cells

The molecular aspects of tigecycline-induced inhibition of cell proliferation were analyzed by the assessment of selected proteins regulating survival and cell proliferation. The levels of MITF, p44/p42 MAPK, and LC3A/B proteins were investigated by Western blotting (Figure 8). In addition, the influence of TGC on p16 and p21 protein expression was evaluated by confocal microscopy imaging (Figure 9).

The obtained results indicated that tigecycline significantly decreased the levels of MITF and p44/p42 MAP kinase in both tested cell lines. In general, the observed effect was stronger in COLO 829 cells. In this case, the levels of MITF and p44/p42 MAPK were reduced by about 99% and 75%, respectively (Figure 8b,d). The corresponding results for A375 cells were about 54% and 23%, respectively (Figure 8a,c). The Western blot analysis showed that tigecycline had a different effect on the LC3A/B protein level, depending on the cell line. The treatment with TGC caused an increase in the LC3A/B level by about 75% in A375 melanoma cells. In turn, the incubation of COLO 829 cells with TGC decreased the protein level by about 59%. In addition, the presented confocal imaging showed that tigecycline increased the expression of the p16 and p21 proteins in tested melanoma cells; however, a significantly stronger effect was noticed for the COLO 829 cell line. The images also demonstrated the influence of the drug on the cell number in tested cultures. It was observed that the treatment with tigecycline caused a significant decrease in the cell population as well as a reduction in the integrity of the colonies formed by the melanoma cells.

## 4. Discussion

Melanoma still remains a challenge for modern medicine. Increasing incidence, the high malignancy of this cancer, and the limited effectiveness of therapy make new solutions in melanoma treatment sought after. One of the directions of the research on new therapies is so-called drug repositioning. The method is based on an attempt to use already registered drugs in new indications. Tetracyclines are among the drugs intensively studied in the context of new pharmacotherapeutic applications. They are broad-spectrum antibiotics that also have many non-antibiotic properties, including anti-inflammatory and anti-cancer activities. The mechanisms of these actions are related to the influence on proteolysis, angiogenesis and apoptosis, among others [30].

The ability of tetracyclines to form complexes with melanin contributes to their accumulation in pigmented tissues [31,32]. This creates a unique opportunity to use tetracyclines in the treatment of skin diseases, including melanoma. Previous studies showed the anti-melanoma potential of doxycycline, minocycline, and chlortetracycline [11,12,33,34]. The obtained results allowed for the conclusion that the activity of tetracyclines against melanoma cells is associated with the inhibition of cell proliferation and the induction of apoptosis.

The results of research showing the anti-melanoma potential of tigecycline were presented in this work. In the first stage, a preliminary evaluation of the cytotoxicity of the drug against amelanotic melanoma A375 and melanotic COLO 829 cells was conducted. The analyses showed that tigecycline proportionally reduced the number of metabolically active melanoma cells to the drug concentration and incubation time. The reduction was already statistically significant from a concentration of 1 µM. It was noticed that the observed decrease was greater in the melenotic type of cells. Taking into consideration the EC_50_ parameter, it was found that tigecycline exhibited stronger activity against melanoma cells compared to normal skin cells, especially melanocytes. The tested drug, at lower concentrations up to 25 µM, did not cause a reduction in the number of metabolically active cells, only in the case of melanocytes. In summary, it can be concluded that the ability of cells to synthesize melanin may have significant implications for the cytotoxicity of tigecycline, as well as the effectiveness and safety of the therapy with the analyzed drug. However, the relative resistance of melanocytes and the greater susceptibility of melanotic melanoma to tigecycline indicate that the roles of melanins and melanogenesis are complex and ambiguous. In this regard, it is important to consider factors such as the type of melanin and its impact on redox homeostasis, cellular metabolism, and inflammation development, as well as drug and toxin binding. The pleiotropic functions and properties of melanin contribute to its involvement in mutagenesis, immunosuppression, and tumor progression [35].

Comparing the calculated EC_50_ values for tigecycline with those obtained for doxycycline [11,33] and minocycline [12,33], it can be concluded that the drugs exhibit a similar effect against melanotic melanoma. The values for 72-h incubation of the drugs with COLO 829 cells were as follows: 16.3 µM for doxycycline, 13.9 µM for minocycline, and 19.2 µM for tigecycline. However, tigecycline proved to be a drug with greater potential against amelanotic melanoma compared to doxycycline and minocycline. In the case of 72-h incubation of A375 cells with doxycycline, minocycline, and tigecycline, the following EC_50_ values were obtained: 110.4 µM, 234.0 µM, and 39.1 µM, respectively.

Apoptosis is a natural, complex, multi-stage, and molecularly controlled process leading to cell death. The dysregulation of apoptosis is considered one of the hallmarks of cancer and is associated with uncontrolled cell proliferation and progression of the disease. Therefore, the induction of apoptosis is one of the beneficial effects of anti-cancer therapy [36,37]. The detection of phosphatidylserine externalization to the outer surface of the lipid bilayer (annexin V binding assay) as well as the analysis of the activity of the executioner caspase 3/7 using FLICA inhibitors were used to assess apoptosis [38,39]. The obtained results revealed that tigecycline increased the percentage of apoptotic cells only in the COLO 829 cell culture. There were no statistically significant differences observed in the A375 cell line. Nevertheless, the cytometric analysis of the tested cultures indicated a slight increase in the percentage of dead cells and the significant inhibition of cell proliferation, regardless of the type of melanoma. The number of A375 and COLO 829 cells practically remained unchanged between 24 and 48 h of incubation with tigecycline. The anti-proliferative effect was also observed in the microscopic images of melanoma cells.

The induction of oxidative stress and disturbances in redox balance in cancer cells constitutes the mechanism of action of many anticancer drugs. This group includes, among others, anthracyclines, platinum coordination complexes, some alkylating agents, camptothecins, and topoisomerase inhibitors [40,41]. The increased production of reactive oxygen species (ROS) can lead to damage and dysfunction of nucleic acids, lipids, enzymes, and many other regulatory proteins [42]. Molecular damage can result in disturbances in proliferation and the induction of cell death [43,44]. The increase in ROS level may be directly related to the drug’s mechanism of action or may have an indirect character, for example, through mitochondrial damage. Mitochondria, as organelles conducting the process of cellular respiration, are a physiological source of ROS. Thus, damage to mitochondrial membranes may contribute not only to the induction of apoptosis but also to an intracellular increase in ROS levels [45,46].

The conducted analyses showed that tigecycline significantly lowered the mitochondrial membrane potential in the examined melanoma cells. It is worth noting that, unlike apoptosis, the effect was observed in both cell lines. This seems significant given the fact that the permeabilization of mitochondrial membranes is one of the stages of the intrinsic apoptotic pathway [47]. At the same time, it was found that tigecycline increased the percentage of A347 and COLO 829 cells with low levels of reduced thiols in a concentration- and time-dependent manner. Reduced forms of thiol compounds, such as glutathione, are part of the intracellular non-enzymatic antioxidant system and are used to neutralize reactive oxygen species [48]. Therefore, it can be assumed that the low level of reduced thiols in melanoma cells indicates tigecycline-induced disturbances in the redox balance. The decrease in MMP during the investigated therapy suggests that mitochondrial damage may have contributed to the induction of oxidative stress. Regardless of the direct cause, the depletion of reduced intracellular thiols can be considered a beneficial effect of tigecycline action. It is believed that the loss of reduced thiols can be a significant element of anticancer therapy. This effect can lead to a gradual accumulation of ROS and the weakening of GSH-mediated detoxification, as well as the improvement of the therapeutic efficacy of ROS-based therapy and decreased drug resistance [49,50].

MITF (microphthalmia-associated transcription factor) is one of the key molecular factors that regulate the differentiation, survival, and proliferation of both normal melanocytes and melanoma cells. An increase in MITF expression in melanoma cells is considered a sign of disease progression, among other things, due to its ability to activate the expression of anti-apoptotic proteins and pro-survival functions [51,52]. It should be noted that the role of MITF in regulating cell division is not straightforward. While a decrease in MITF only moderately suppresses cell proliferation, high levels of transcription factor are observed in more proliferative and less invasive melanoma cells [53,54]. However, it is also worth emphasizing that the activity of MITF can be increased through phosphorylation involving extracellular signal-regulated kinases 1/2 (Erk1/2), also known as p44/42 MAPK [55,56]. In general, the high expression and activity of p44/42 MAP kinases favor cell survival and proliferation and may lead to tumor development, the progression and inhibition of apoptosis, and even drug-stimulated [57,58,59].

The obtained results showed that tigecycline decreased the levels of MITF and p44/42 MAPK proteins in both tested melanoma cell lines. The observed decrease was significantly greater in the melanotic melanoma cells than in the amelanotic type of cells. The level of MITF in COLO 829 cells was reduced to about 1% when compared to the control. These changes indicate the anti-proliferative effect of tigecycline on melanoma cells can be related to the reduced levels of MITF and p44/42 MAP kinases. The noted decreases in MITF and p44/42 MAPK in individual melanoma cell lines also reflect the greater sensitivity of COLO 829 cells to the action of tigecycline, as demonstrated in other analyses.

The difference in the levels of LC3A/B between the COLO 829 and A375 cell lines also appears to be extremely significant. This protein is the molecular marker of autophagy, a physiological process that leads to the degradation of damaged proteins and organelles during stressful conditions [60]. Regarding cancer cells, it should be acknowledged that the role of autophagy is not unequivocal. This process is considered both a promoter and a suppressor of tumors. On the one hand, autophagy can protect cells from malignant transformation, among other things, by limiting the harmful effects of ROS and by preventing DNA damage. On the other hand, the process contributes to the growth and survival of tumor cells by recycling macromolecules and supplying essential substrates under unfavorable conditions [61]. Thus, it was concluded that autophagy manipulation may either promote or hinder the growth and development of cancer. The final result could depend on the origin of the cancer cells, the type of cancer, and the response to treatment [62]. Moreover, it was found that autophagy can inhibit apoptosis by selectively reducing the abundance of pro-apoptotic proteins, e.g., active caspase-8 [63,64]. The above considerations suggest that the lack of apoptotic cells in the A375 melanoma culture may result from the induction of autophagy by tigecycline. In this aspect, autophagy may be an undesirable process that limits the effectiveness of therapy. Importantly, such an effect was not observed in COLO 829 cells. Tigecycline in melanotic melanoma cells reduced the levels of LC3A/B and triggered the process of apoptosis.

The anti-proliferative effect of tigecycline was also evaluated by analyzing the cell cycle. The conducted studies showed that the profile of changes in the individual phases of the cycle after the treatment with tigecycline differs depending on the melanoma cell line. While in A375 cells, tigecycline caused a decrease in the percentage of cells in the G1/G0 phase and an increase in the number of cells in the S and G2/M phases, in COLO 829 culture, a cell accumulation in the G1/G0 phase as well as a decrease in the number of cells in the remaining phases were observed. It should be noted that the cell cycle changes induced by tigecycline in melanotic melanoma cells are consistent with the analysis of p21 and p16 protein expression using confocal microscopy. Both proteins belong to cyclin-dependent kinase inhibitors and are responsible for G1/G0 phase cell cycle arrest [65]. The presented images indicated that a significant increase in the levels of p21 and p16 only occurred in COLO 829 cells. Only a slight increase in the investigated proteins was observed compared to the control in A375 melanoma cells. It is worth emphasizing that earlier studies on melanoma cells showed that the inhibition of p44/42 MAP kinase may lead to an increase in the expression of p21 and p16 proteins [66]. Therefore, it can be assumed that a significant decrease in the level of p44/42 MAPK in COLO 829 cells after the treatment contributes to the increase in the expression of p16 and p21 and the arrest of the cell cycle in the G1/G0 phase.

The presented results indicate the therapeutic potential of tigecycline in the treatment of melanoma. The observed differences between melanoma types may be due to the involvement of melanins and melanogenesis in the cytotoxic effect of the drug. However, it should be taken into account that the tested cell lines were cultured in different media, which could have influenced the final outcome to some extent. Further studies confirming the effectiveness of tigecycline using other melanoma cell lines appear necessary. It is important to consider the genetic background, the phenotypic characteristics of the cells, and a detailed analysis of the melanogenesis process. Conducting in vivo studies is also crucial, especially considering the significant interaction between the neuroendocrine mechanisms regulating body homeostasis and melanoma cells [67]. It is also important to consider the antibiotic activity of tigecycline. This action can be significant for the effectiveness and safety of therapy in in vivo conditions. Furthermore, the spreading and increasing bacterial resistance to antibiotics, including tetracyclines, pose certain limitations for utilizing the non-antibiotic properties of tigecycline. Nevertheless, in justified cases, controlled treatment with tigecycline appears to be feasible.

## 5. Conclusions

In summary, the obtained results indicate that tigecycline has therapeutic potential for melanotic and amelanotic melanoma cells. The main observed effect in both melanoma cell lines was anti-proliferative action, which was proportional to the drug concentration and incubation time. The inhibition of cell division was associated with an increase in the number of cells with low levels of reduced thiols, a decrease in mitochondrial potential, and a decrease in MITF and p44/42 MAPK levels. However, it should be noted that specific differences in drug action on individual cell lines were observed. COLO 829 cells were more sensitive to tigecycline. The drug showed a stronger effect on melanotic melanoma cells by inducing apoptosis and inhibiting the cell cycle in the G1/G0 phase, which was probably due to an increase in p21 and p16 protein levels. A375 cells were more resistant to tigecycline. The treatment did not induce programmed cell death but contributed to an increase in LC3A/B protein levels—an autophagy marker. The presented results certainly contribute to expanding the knowledge of tetracycline pharmacology and also enable a better understanding of melanoma cell biology. They also draw attention to the significant differences between different lines of the same type of cancer.

## Figures and Tables

**Figure 1 cells-12-01564-f001:**
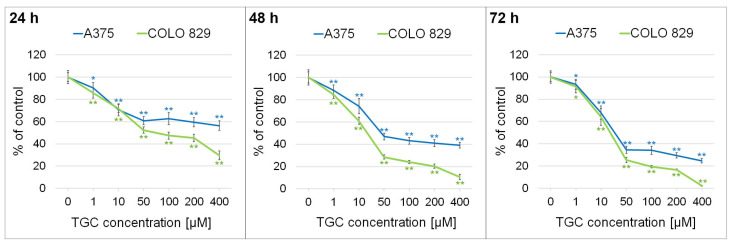
The general assessment of tigecycline cytotoxic potential on A375 and COLO 829 cell lines. Melanoma cells were incubated with the analyzed drug for 24 h, 48 h, and 72 h. The results were expressed as a percentage of control. Mean values ± SD were presented. * *p* < 0.05; ** *p* < 0.005.

**Figure 2 cells-12-01564-f002:**
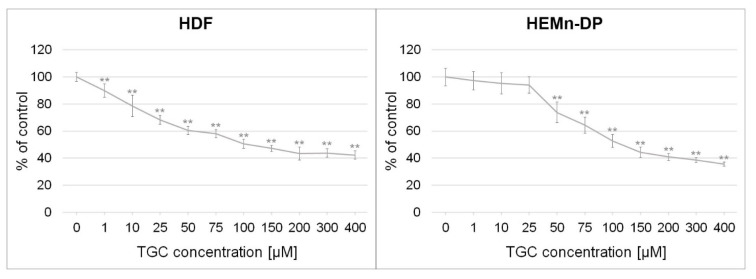
The analysis of tigecycline cytotoxic potential on human dermal fibroblasts (HDF) and human epidermal melanocytes (HEMn-DP). The cells were treated for 48 h. The results were expressed as a percentage of control. Mean values ± SD were presented. ** *p* < 0.005.

**Figure 3 cells-12-01564-f003:**
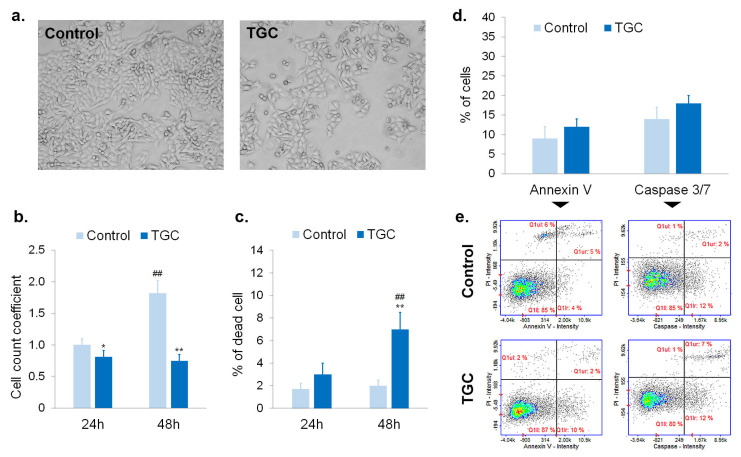
The evaluation of cell proliferation and apoptosis of A375 melanoma cells treated with 400 µM of tigecycline—(**a**) microscopic analysis of cell morphology after 48-h incubation with tigecycline; (**b**) analysis of cell number and (**c**) cell viability in tested cultures; analysis of apoptosis expressed as (**d**) the mean values of percentage of annexin V-positive cells and cells with activated caspase 3/7 after 48-h treatment as well as (**e**) representative scatter plot. * *p* < 0.05; ** *p* < 0.005 vs. control; ## *p* < 0.005 vs. corresponding 24 h sample.

**Figure 4 cells-12-01564-f004:**
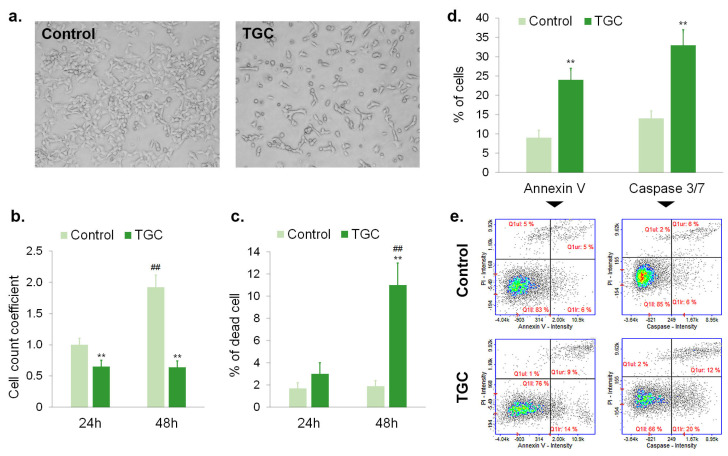
The evaluation of cell proliferation and apoptosis of COLO 829 melanoma cells treated with 200 µM of tigecycline—(**a**) microscopic analysis of cell morphology after 48-h incubation with tigecycline; (**b**) analysis of cell number and (**c**) cell viability in tested cultures; analysis of apoptosis expressed as (**d**) the mean values of percentage of annexin V-positive cells and cells with activated caspase 3/7 after 48-h treatment as well as (**e**) representative scatter plot. ** *p* < 0.005 vs. control; ## *p* < 0.005 vs. corresponding 24 h sample.

**Figure 5 cells-12-01564-f005:**
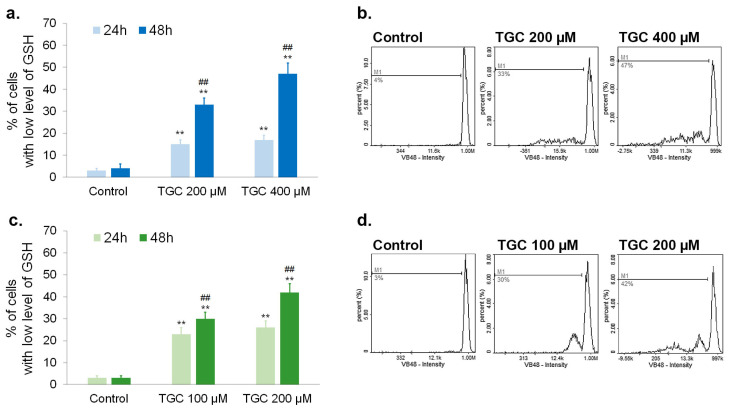
The analysis of reduced thiols in melanoma cells after the treatment with tigecycline. The results were presented as the mean values and representative histograms for A375 (**a**,**b**) and COLO 829 cell lines (**c**,**d**). ** *p* < 0.005 vs. control; ## *p* < 0.005 vs. corresponding 24 h sample.

**Figure 6 cells-12-01564-f006:**
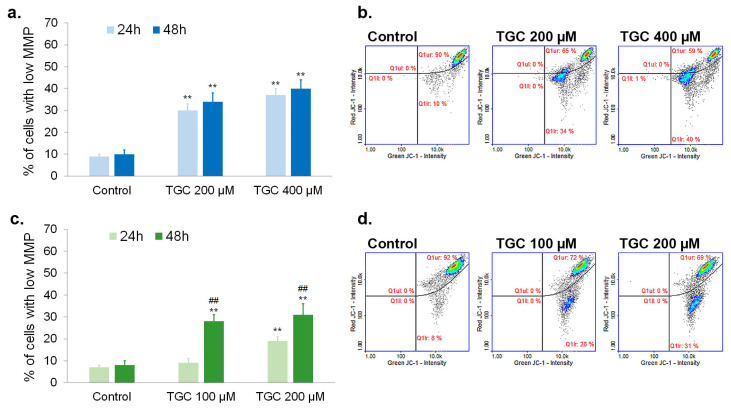
The analysis of mitochondrial membrane potential in melanoma cells after the treatment with tigecycline. The mean values as well as representative scatter plots were presented for A375 (**a**,**b**) and COLO 829 cell lines (**c**,**d**). ** *p* < 0.005 vs. control; ## *p* < 0.005 vs. corresponding 24 h sample.

**Figure 7 cells-12-01564-f007:**
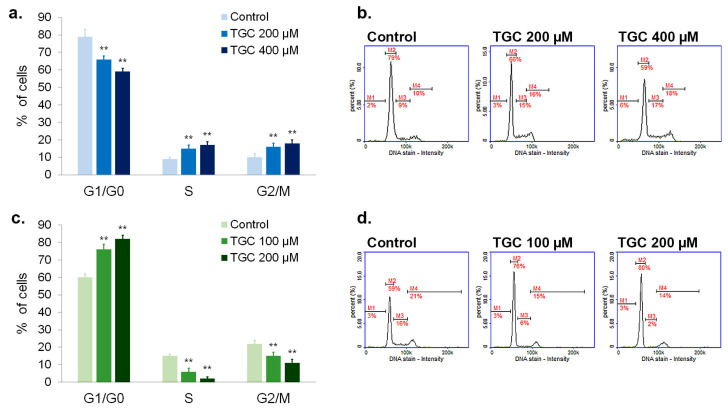
The analysis of cell cycle after the 48-h treatment with tigecycline. The mean values as well as representative histograms were presented for A375 (**a**,**b**) and COLO 829 cell lines (**c**,**d**). ** *p* < 0.005 vs. control.

**Figure 8 cells-12-01564-f008:**
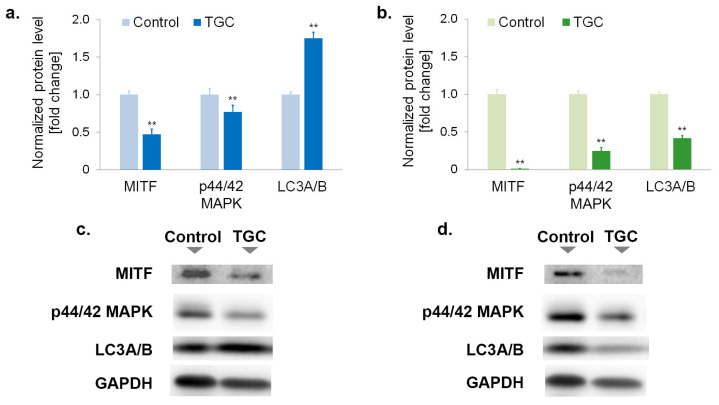
Western blot analysis of MITF, p44/p42 MAPK, and LC3A/B proteins in melanoma cells after 48-h treatment. Bar graphs show normalized results for A375 cells treated with 400 µM of tigecycline (**a**), and COLO 829 cells incubated with the drug in a concentration of 200 µM (**b**). Corresponding representative blot images (**c**,**d**) are also presented. ** *p* < 0.005 vs. control.

**Figure 9 cells-12-01564-f009:**
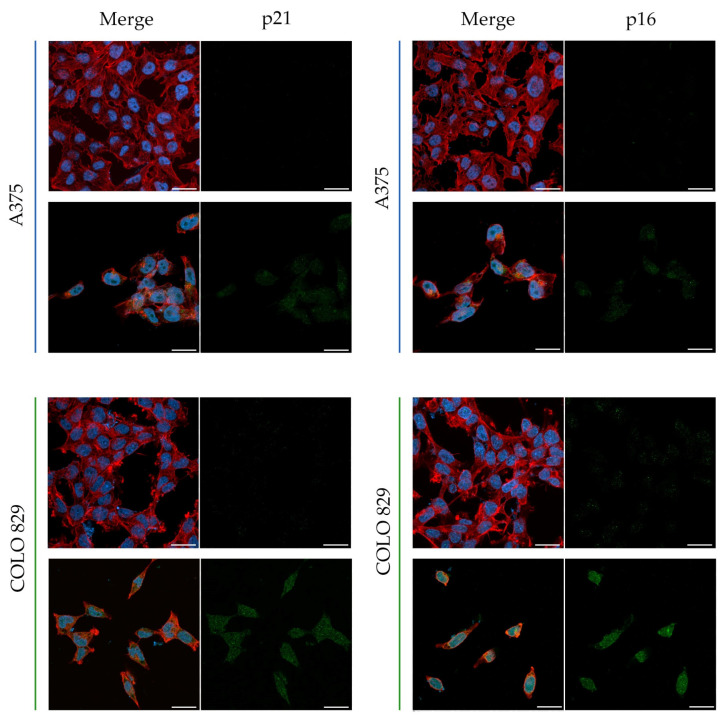
Scanning confocal microscopy imaging presenting the expression of p16 and p21 proteins in melanoma cells after 48-h incubation with tigecycline. A375 cells were treated with 400 µM of tigecycline, and COLO 829 cells were incubated with the drug in a concentration of 200 µM. Scale bar 20 µm.

## Data Availability

The data presented in this study are available upon request from the corresponding author.

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
