# Peer review of "The Assessment of Anti-Melanoma Potential of Tigecycline—Cellular and Molecular Studies of Cell Proliferation, Apoptosis and Autophagy on Amelanotic and Melanotic Melanoma Cells"

_cells, 2023, doi:10.3390/cells12121564_

Round 1
Reviewer 1 Report
The topic of developing anti-melanoma drugs is of interest. The methodology, data collection and their interpretation appear to be fine. However, the paper will benefit for the corrections amendments.
The cell biology technique used appear to be appropriately described.
However, better description of cell lines is necessary.
It is said that COLO 829 is melanotic while A375 is amelanotic. Presentation of cell pellets and higher mag of cells in culture would be appreciated by readers to see relative level of pigmentation and morphology. A375 has epithelioid morphology. Note, that in the description of ATCC morphology of COLO 829 is listed as fibroblast.
In addition, the mutation status for gene of interest in melanoma could be provided, if known. For example A375 is NRAS mutant. What about BRAF?
Since authors mention melanotic and amelanotic phenotype of the cells, I am surprised by lack of even mentioning that melanin and melanogenesis can affect melanoma behavior, responsiveness to therapy, etc as discussed recently: Frontiers in Oncology 2022;12. DOI: 10.3389/fonc.2022.842496. If fact melanogenesis can affect cellular metabolism in melanoma, responsiveness to therapy and suppress immune responses. Also there is clinico-pathological data showing poorer outcome for pigmented advanced melanoma vs non-pigmented. These issues deserve brief mentioning.
As relates to cell lines. Thye were cultured in different media, and the media by themselves could affect the responses. This could be mentioned.
Also to suggest that melanotic is more responsive than amelanotic would require lines with both phenotype from the same genetic background, or simple inhibition of melanogenesis in melanotic and comparing the drug effect. This could be mentioned in limitations.
I also suggest adding in limitation section that the efficiency of proposed drugs remain to be tested in vivo for example in nude mice, and more melanoma lines could be tested to see how general the effect is. While you mention deadly behavior of melanoma tumor, the readers would appreciate to learn that advanced melanomas can affect body homeostasis: How cancer hijacks the body’s homeostasis through the neuroendocrine system. Trends Neurosci 46: 263-275, 2023.
The English is fine, minor editing of English language would be required
Author Response
Response to the Reviewer #1
Dear Reviewer,
We are grateful for a thorough and insightful assessment of our manuscript. The provided comments and advice are accurate and valuable. Having taken into consideration the remarks, we have carefully revised and corrected the manuscript. All edited fragments of the manuscript have been marked in red. We hope that the improved version of the paper will be satisfying for both the Reviewer and the Editors.
According to the Reviewer's suggestion, we have added the key information characterizing the melanoma cell lines used in our study. Unfortunately, we currently do not possess photographs of cell pellets to include in the publication. However, in one of our previous publications, we demonstrated relatively high expression of tyrosinase in COLO 829 cells using confocal microscopy, indicating a melanotic type and the potential for melanin synthesis (https://doi.org/10.3390/ijms21186917). Our future studies aim to elucidate the role of melanin and the melanogenesis process in terms of resistance and the effectiveness of melanoma treatment. We will conduct further detailed analyses using different in vitro experimental models, including additional supplementation of melanin synthesis substrates and melanogenesis inhibitors. We will definitely take advantage of the valuable advice from the Reviewer and take photos of the cell pellets.
Following the reviewer's suggestion, we have addressed the issue of melanogenesis and melanin in the discussion section. This topic has also been discussed in relation to the presented results of the preliminary analysis of tigecycline cytotoxicity on human epidermal melanocytes and dermal fibroblasts:
„The results of research showing the anti-melanoma potential of tigecycline were presented in this work. In the first stage, a preliminary evaluation of the cytotoxicity of the drug against amelanotic melanoma A375 and melanotic COLO 829 cells was conducted. The analyses showed that tigecycline reduced the number of metabolically active melanoma cells proportionally to the drug concentration and incubation time. The reduction was statistically significant already from a concentration of 1 µM. It was noticed that the observed decrease was greater in the melenotic type of cells. Taking into consideration the EC50 parameter, it was found that tigecycline exhibited stronger activity against melanoma cells compared to normal skin cells, especially melanocytes. The tested drug, at lower concentrations, up to 25 µM, did not caused a reduction in the number of metabolically active cells only in the case of melanocytes. In summary, it can be concluded that the ability of cells to synthesize melanin may have significant implications for the cytotoxicity of tigecycline, as well as the effectiveness and safety of the therapy with the analyzed drug. However, the relative resistance of melanocytes and the greater susceptibility of melanotic melanoma to tigecycline indicate that the role of melanins and melanogenesis is complex and ambiguous. In this regard, it is important to consider factors such as the type of melanin and its impact on redox homeostasis, cellular metabolism, inflammation development, as well as drug and toxin binding. The pleiotropic functions and properties of melanin contribute to its involvement in mutagenesis, immunosuppression, and tumor progression [35].”
- Slominski, R.M.; Sarna, T.; Płonka, P.M.; Raman, C.; Brożyna, A.A.; Slominski, A.T. Melanoma, Melanin, and Melanogenesis: The Yin and Yang Relationship. Front. Oncol. 2022, 12, 842496. doi:10.3389/fonc.2022.842496
Taking into account the valuable comments from the Reviewer, we have added a section at the end of the discussion addressing certain limitations of the study and highlighting important issues that need to be addressed in future stages of evaluating the action of tigecycline:
“The presented results indicate the therapeutic potential of tigecycline in the treatment of melanoma. The observed differences between melanoma types may be due to the involvement of melanins and melanogenesis in the cytotoxic effect of the drug. However, it should be taken into account that the tested cell lines were cultured in different media, which could have influenced the final outcome to some extent. Further studies confirming the effectiveness of tigecycline using other melanoma cell lines appear necessary. It is important to consider the genetic background, phenotypic characteristics of the cells, and a detailed analysis of the melanogenesis process. Conducting in vivo studies is also crucial, especially considering the significant interaction between neuroendocrine mechanisms regulating body homeostasis and melanoma cells [67]. It is also important to consider the antibiotic activity of tigecycline. This action can be significant for the effectiveness and safety of therapy in in vivo conditions. Furthermore, the spreading and increasing bacterial resistance to antibiotics, including tetracyclines, pose certain limitations for utilizing the non-antibiotic properties of tigecycline. Nevertheless, in justified cases, controlled treatment with tigecycline appears to be feasible.”
- Slominski, R.M.; Raman, C.; Chen, J.Y.; Slominski, A.T. How cancer hijacks the body's homeostasis through the neuroendocrine system. Trends Neurosci. 2023, 46, 263-275. doi:10.1016/j.tins.2023.01.003
Reviewer 2 Report
I read this article with great interest. It brings a new approach to the therapy of malignant melanoma. The use of antibiotics in a pathology other than the infectious one can be a milestone not only for malignant melanoma but also for other types of cancer.
In order to increase the value of this article, I think that in the Introduction chapter it is necessary to present the cellular mechanisms induced by the studied antibiotic treatment.
minor revision
Author Response
Response to the Reviewer #2
Dear Reviewer,
We would like to express our gratitude for proofreading and evaluating our article, as well as for your kind opinion. We completely agree with the Reviewer that incorporating additional information about the molecular mechanisms will enhance the quality and clarity of the publication. Following the Reviewer's suggestion, we have included new information about the action of tetracyclines in the article. We believe that the revised version of the publication will meet the Reviewer's expectations.
“According to previous scientific reports, tetracycline antibiotics, such as doxycycline and minocycline, have shown promising therapeutic effects in cancer treatment, including melanoma [11, 12]. The molecular mechanisms of their anticancer action involve, among others, the inhibition of matrix metalloproteinase, HSP90, as well as the ERK/MAPK and PAR1/FAK/PI3K/AKT pathways. They also increase the levels of the proapoptotic protein BAX and decrease the levels of the anti-apoptotic protein BCL-2. These antibiotics reduce also proliferation markers such as Ki67 and PCNA, as well as the tumor necrosis factor TNF-α, by inhibiting the activation of NF-κB [13-18]. As a result, these drugs can arrest cell proliferation, induce apoptosis, and inhibit cell migration and the development cancer stem cells. Furthermore, it has been demonstrated that doxycycline and minocycline inhibit angiogenesis by affecting IL-8 and translational suppression of HIF-1α, respectively [19, 20].
Tigecycline, a third-generation tetracycline that includes a tert-butylglycylamide substituent at the C-9 position, has been found to have both antibiotic and non-antibiotic properties and, among other things, anticancer activity [21]. The drug was approved for use in medicine in 2005 for complicated and resistant bacterial infections due to its extended spectrum antibiotic properties [22, 23]. Its bacteriostatic activity results from the ability to inhibit bacterial protein synthesis by binding to the 30S subunit of the bacterial ribosome [24]. Moreover, there are reports indicating that tigecycline may have potential in the treatment of various types of cancer, including ovarian, breast, cervical, thyroid, lung, pancreatic, stomach, and skin cancers [16, 25]. The main anticancer effects of tigecycline are similar to those observed for doxycycline and minocycline. These effects involve the inhibition of cell proliferation, induction of apoptosis, influence on autophagy activation, blockade of angiogenesis, and suppression of migration. Molecular mechanisms underlying these actions include significant inhibition of mitochondrial translation and signaling pathways related to p21CIP1/Waf1, Wnt/β, Akt, and mTOR kinase [26, 27, 28]. A previously conducted study on melanoma cells demonstrated that tigecycline prevented epithelial-mesenchymal transition and arrested the cell cycle at the G0/G1 phase. These actions were dependent on the cyclin-dependent kinase inhibitor p21CIP1/Waf1 [29].”
- Rok, J.; Karkoszka, M.; Rzepka, Z.; Respondek, M.; Banach, K.; Beberok, A.; Wrześniok, D. Cytotoxic and proapoptotic effect of doxycycline - An in vitro study on the human skin melanoma cells. Toxicol. In Vitro. 2020, 65, 104790. doi:10.1016/j.tiv.2020.104790
- Rok, J.; Rzepka, Z.; Beberok, A.; Pawlik, J.; Wrześniok, D. Cellular and Molecular Aspects of Anti-Melanoma Effect of Minocycline-A Study of Cytotoxicity and Apoptosis on Human Melanotic Melanoma Cells. Int. J. Mol. Sci. 2020, 21, 6917. doi: 10.3390/ijms21186917
- Markowska, A.; Kaysiewicz, J.; Markowska, J.; Huczyński, A. Doxycycline, salinomycin, monensin and ivermectin repositioned as cancer drugs. Bioorg. Med. Chem. Lett. 2019, 29, 1549-1554. doi:10.1016/j.bmcl.2019.04.045
- Hadjimichael, A.C.; Foukas, A.F.; Savvidou, O.D.; Mavrogenis, A.F.; Psyrri, A.K.; Papagelopoulos, P.J. The anti-neoplastic effect of doxycycline in osteosarcoma as a metalloproteinase (MMP) inhibitor: a systematic review. Clin. Sarcoma Res. 2020, 10, 7. doi:10.1186/s13569-020-00128-6
- Afshari, A.R.; Mollazadeh, H.; Sahebkar, A. Minocycline in Treating Glioblastoma Multiforme: Far beyond a Conventional Antibiotic. J. Oncol. 2020, 2020, 8659802. doi:10.1155/2020/8659802
- Araújo, D.; Ribeiro, E.; Amorim, I.; Vale, N. Repurposed Drugs in Gastric Cancer. Molecules 2022, 28, 319. doi:10.3390/molecules28010319
- Liu, H.; Tao, H.; Wang, H.; Yang, Y.; Yang, R.; Dai, X.; Ding, X.; Wu, H.; Chen, S.; Sun, T. Doxycycline Inhibits Cancer Stem Cell-Like Properties via PAR1/FAK/PI3K/AKT Pathway in Pancreatic Cancer. Front. Oncol. 2021, 10, 619317. doi: 10.3389/fonc.2020.619317
- Ali, I.; Alfarouk, K.O.; Reshkin, S.J.; Ibrahim, M.E. Doxycycline as Potential Anti-cancer Agent. Anticancer Agents Med. Chem. 2017, 17, 1617-1623. doi: 10.2174/1871520617666170213111951
- Ghasemi, K.; Ghasemi, K. A Brief look at antitumor effects of doxycycline in the treatment of colorectal cancer and combination therapies. Eur J Pharmacol. 2022, 916, 174593. doi:10.1016/j.ejphar.2021.174593
- Jung, H.J.; Seo, I.; Jha, B.K.; Suh, S.I.; Suh, M.H.; Baek, W.K. Minocycline inhibits angiogenesis in vitro through the translational suppression of HIF-1α. Arch. Biochem. Biophys. 2014, 545, 74-82. doi:10.1016/j.abb.2013.12.023
- Zhanel, G.G.; Karlowsky, J.A.; Rubinstein, E.; Hoban, D.J. Tigecycline: a novel glycylcycline antibiotic. Expert Rev. Anti. Infect. Ther. 2006, 4, 9-25. doi:10.1586/14787210.4.1.9
- Meagher, A.K.; Ambrose, P.G.; Grasela, T.H.; Ellis-Grosse, E.J. Pharmacokinetic/pharmacodynamic profile for tigecycline-a new glycylcycline antimicrobial agent. Diagn. Microbiol. Infect. Dis. 2005, 52, 165-171. doi:10.1016/j.diagmicrobio.2005.05.006
- Yaghoubi, S.; Zekiy, A.O.; Krutova, M.; Gholami, M.; Kouhsari, E.; Sholeh, M.; Ghafouri, Z.; Maleki, F. Tigecycline antibacterial activity, clinical effectiveness, and mechanisms and epidemiology of resistance: narrative review. Eur. J. Clin. Micro-biol. Infect. Dis. 2022, 41, 1003-1022. doi: 10.1007/s10096-020-04121-1
- Rose, W.E.; Rybak, M.J. Tigecycline: first of a new class of antimicrobial agents. Pharmacotherapy 2006, 26, 1099-1110. doi: 10.1592/phco.26.8.1099. PMID: 16863487
- Skrtić, M.; Sriskanthadevan, S.; Jhas, B.; Gebbia, M.; Wang, X.; Wang, Z.; Hurren, R.; Jitkova, Y.; Gronda, M.; Maclean, N.; Lai, C,K.; Eberhard, Y.; Bartoszko, J.; Spagnuolo, P.; Rutledge, A.C.; Datti, A.; Ketela, T.; Moffat, J.; Robinson, B.H.; Cameron, J.H.; Wrana, J.; Eaves, C.J.; Minden, M.D.; Wang, J.C.; Dick, J.E.; Humphries, K.; Nislow, C.; Giaever, G.; Schimmer, A.D. Inhibition of mitochondrial translation as a therapeutic strategy for human acute myeloid leukemia. Cancer Cell. 2011, 20, 674-88. doi: 10.1016/j.ccr.2011.10.015
- Dong, Z.; Abbas, M.N.; Kausar, S.; Yang, J.; Li, L.; Tan, L.; Cui, H. Biological Functions and Molecular Mechanisms of Antibiotic Tigecycline in the Treatment of Cancers. Int. J. Mol. Sci. 2019, 20, 3577. doi:10.3390/ijms20143577
- Xu, Z.; Yan, Y.; Li, Z.; Qian, L.; Gong, Z. The Antibiotic Drug Tigecycline: A Focus on its Promising Anticancer Properties. Front. Pharmacol. 2016, 7, 473. doi:10.3389/fphar.2016.00473
- Tang, C.; Yang, L.; Jiang, X.; Xu, C.; Wang, M.; Wang, Q.; Zhou, Z.; Xiang, Z.; Cui, H. Antibiotic drug tigecycline inhibited cell proliferation and induced autophagy in gastric cancer cells. Biochem. Biophys. Res. Commun. 2014, 446, 105-112. doi: 10.1016/j.bbrc.2014.02.043
- Hu, H.; Dong, Z.; Tan, P.; Zhang, Y.; Liu, L.; Yang, L.; Liu, Y.; Cui, H. Antibiotic drug tigecycline inhibits melanoma progression and metastasis in a p21CIP1/Waf1-dependent manner. Oncotarget 2016, 7, 3171-3185. doi: 10.18632/oncotarget.6419
Reviewer 3 Report
In this manuscript, Rok et al assessed the cytotoxic effects of tigecycline on melanoma cell lines A375 and COLO829. This study demonstrated that tigecycline inhibited cell proliferation, associated with cell cycle dysregulation, depolarization of mitochondrial membrane and induced apoptosis in COLO829 cells. This manuscript is very interesting, well written with well defined material and methods. I just have some minor observations that I would like the authors to consider.
1. Introduction - Consider expanding on the mechanism of action and anticancer effects of doxycycline and minocycline, also to address here or on the discussion the rationale of using both melanotic and amelanotic cell lines.
2. Results - Has the effect of tigecycline on normal cells been assessed, or how specific is this approach to cancer cells, this can also be addressed on the discussion
3. Results - Line 221: it says hand instead of the abbreviation "h" for hour, Line 243 it says " a lot of" I would recommend rephrasing this and possible add percentage of cells showing that morphology
4. Discussion - Another point that can be addressed are the potential challenges of using an antibiotic such as tigecyline (which is typically reserved for difficult to treat infections) for other uses such, due to the fact of the emerging threat of antibiotic multi resistance
I believe this manuscript is very well written and only requires minor editing changes that I mentioned above
Author Response
Response to the Reviewer #3
Dear Reviewer,
We would like to express our gratitude for reviewing our research findings and for your positive opinion. Furthermore, we appreciate the guidance and suggestions provided, all of which we fully agree with. In light of the review, we have made revisions to the manuscript based on the Reviewer's comments. We hope that the new version of the publication will be satisfactory and meet the expectations of the Reviewer.
Ad. 1. In the Introduction section, we have introduced information regarding the mechanisms of action of doxycycline, minocycline, and tigecycline.
“According to previous scientific reports, tetracycline antibiotics, such as doxycycline and minocycline, have shown promising therapeutic effects in cancer treatment, including melanoma [11, 12]. The molecular mechanisms of their anticancer action involve, among others, the inhibition of matrix metalloproteinase, HSP90, as well as the ERK/MAPK and PAR1/FAK/PI3K/AKT pathways. They also increase the levels of the proapoptotic protein BAX and decrease the levels of the anti-apoptotic protein BCL-2. These antibiotics reduce also proliferation markers such as Ki67 and PCNA, as well as the tumor necrosis factor TNF-α, by inhibiting the activation of NF-κB [13-18]. As a result, these drugs can arrest cell proliferation, induce apoptosis, and inhibit cell migration and the development cancer stem cells. Furthermore, it has been demonstrated that doxycycline and minocycline inhibit angiogenesis by affecting IL-8 and translational suppression of HIF-1α, respectively [19, 20].
Tigecycline, a third-generation tetracycline that includes a tert-butylglycylamide substituent at the C-9 position, has been found to have both antibiotic and non-antibiotic properties and, among other things, anticancer activity [21]. The drug was approved for use in medicine in 2005 for complicated and resistant bacterial infections due to its extended spectrum antibiotic properties [22, 23]. Its bacteriostatic activity results from the ability to inhibit bacterial protein synthesis by binding to the 30S subunit of the bacterial ribosome [24]. Moreover, there are reports indicating that tigecycline may have potential in the treatment of various types of cancer, including ovarian, breast, cervical, thyroid, lung, pancreatic, stomach, and skin cancers [16, 25]. The main anticancer effects of tigecycline are similar to those observed for doxycycline and minocycline. These effects involve the inhibition of cell proliferation, induction of apoptosis, influence on autophagy activation, blockade of angiogenesis, and suppression of migration. Molecular mechanisms underlying these actions include significant inhibition of mitochondrial translation and signaling pathways related to p21CIP1/Waf1, Wnt/β, Akt, and mTOR kinase [26, 27, 28]. A previously conducted study on melanoma cells demonstrated that tigecycline prevented epithelial-mesenchymal transition and arrested the cell cycle at the G0/G1 phase. These actions were dependent on the cyclin-dependent kinase inhibitor p21CIP1/Waf1 [29].”
- Rok, J.; Karkoszka, M.; Rzepka, Z.; Respondek, M.; Banach, K.; Beberok, A.; Wrześniok, D. Cytotoxic and proapoptotic effect of doxycycline - An in vitro study on the human skin melanoma cells. Toxicol. In Vitro. 2020, 65, 104790. doi:10.1016/j.tiv.2020.104790
- Rok, J.; Rzepka, Z.; Beberok, A.; Pawlik, J.; Wrześniok, D. Cellular and Molecular Aspects of Anti-Melanoma Effect of Minocycline-A Study of Cytotoxicity and Apoptosis on Human Melanotic Melanoma Cells. Int. J. Mol. Sci. 2020, 21, 6917. doi: 10.3390/ijms21186917
- Markowska, A.; Kaysiewicz, J.; Markowska, J.; Huczyński, A. Doxycycline, salinomycin, monensin and ivermectin repositioned as cancer drugs. Bioorg. Med. Chem. Lett. 2019, 29, 1549-1554. doi:10.1016/j.bmcl.2019.04.045
- Hadjimichael, A.C.; Foukas, A.F.; Savvidou, O.D.; Mavrogenis, A.F.; Psyrri, A.K.; Papagelopoulos, P.J. The anti-neoplastic effect of doxycycline in osteosarcoma as a metalloproteinase (MMP) inhibitor: a systematic review. Clin. Sarcoma Res. 2020, 10, 7. doi:10.1186/s13569-020-00128-6
- Afshari, A.R.; Mollazadeh, H.; Sahebkar, A. Minocycline in Treating Glioblastoma Multiforme: Far beyond a Conventional Antibiotic. J. Oncol. 2020, 2020, 8659802. doi:10.1155/2020/8659802
- Araújo, D.; Ribeiro, E.; Amorim, I.; Vale, N. Repurposed Drugs in Gastric Cancer. Molecules 2022, 28, 319. doi:10.3390/molecules28010319
- Liu, H.; Tao, H.; Wang, H.; Yang, Y.; Yang, R.; Dai, X.; Ding, X.; Wu, H.; Chen, S.; Sun, T. Doxycycline Inhibits Cancer Stem Cell-Like Properties via PAR1/FAK/PI3K/AKT Pathway in Pancreatic Cancer. Front. Oncol. 2021, 10, 619317. doi: 10.3389/fonc.2020.619317
- Ali, I.; Alfarouk, K.O.; Reshkin, S.J.; Ibrahim, M.E. Doxycycline as Potential Anti-cancer Agent. Anticancer Agents Med. Chem. 2017, 17, 1617-1623. doi: 10.2174/1871520617666170213111951
- Ghasemi, K.; Ghasemi, K. A Brief look at antitumor effects of doxycycline in the treatment of colorectal cancer and combination therapies. Eur J Pharmacol. 2022, 916, 174593. doi:10.1016/j.ejphar.2021.174593
- Jung, H.J.; Seo, I.; Jha, B.K.; Suh, S.I.; Suh, M.H.; Baek, W.K. Minocycline inhibits angiogenesis in vitro through the translational suppression of HIF-1α. Arch. Biochem. Biophys. 2014, 545, 74-82. doi:10.1016/j.abb.2013.12.023
- Zhanel, G.G.; Karlowsky, J.A.; Rubinstein, E.; Hoban, D.J. Tigecycline: a novel glycylcycline antibiotic. Expert Rev. Anti. Infect. Ther. 2006, 4, 9-25. doi:10.1586/14787210.4.1.9
- Meagher, A.K.; Ambrose, P.G.; Grasela, T.H.; Ellis-Grosse, E.J. Pharmacokinetic/pharmacodynamic profile for tigecycline-a new glycylcycline antimicrobial agent. Diagn. Microbiol. Infect. Dis. 2005, 52, 165-171. doi:10.1016/j.diagmicrobio.2005.05.006
- Yaghoubi, S.; Zekiy, A.O.; Krutova, M.; Gholami, M.; Kouhsari, E.; Sholeh, M.; Ghafouri, Z.; Maleki, F. Tigecycline antibacterial activity, clinical effectiveness, and mechanisms and epidemiology of resistance: narrative review. Eur. J. Clin. Microbiol. Infect. Dis. 2022, 41, 1003-1022. doi: 10.1007/s10096-020-04121-1
- Rose, W.E.; Rybak, M.J. Tigecycline: first of a new class of antimicrobial agents. Pharmacotherapy 2006, 26, 1099-1110. doi: 10.1592/phco.26.8.1099. PMID: 16863487
- Skrtić, M.; Sriskanthadevan, S.; Jhas, B.; Gebbia, M.; Wang, X.; Wang, Z.; Hurren, R.; Jitkova, Y.; Gronda, M.; Maclean, N.; Lai, C,K.; Eberhard, Y.; Bartoszko, J.; Spagnuolo, P.; Rutledge, A.C.; Datti, A.; Ketela, T.; Moffat, J.; Robinson, B.H.; Cameron, J.H.; Wrana, J.; Eaves, C.J.; Minden, M.D.; Wang, J.C.; Dick, J.E.; Humphries, K.; Nislow, C.; Giaever, G.; Schimmer, A.D. Inhibition of mitochondrial translation as a therapeutic strategy for human acute myeloid leukemia. Cancer Cell. 2011, 20, 674-88. doi: 10.1016/j.ccr.2011.10.015
- Dong, Z.; Abbas, M.N.; Kausar, S.; Yang, J.; Li, L.; Tan, L.; Cui, H. Biological Functions and Molecular Mechanisms of Antibiotic Tigecycline in the Treatment of Cancers. Int. J. Mol. Sci. 2019, 20, 3577. doi:10.3390/ijms20143577
- Xu, Z.; Yan, Y.; Li, Z.; Qian, L.; Gong, Z. The Antibiotic Drug Tigecycline: A Focus on its Promising Anticancer Properties. Front. Pharmacol. 2016, 7, 473. doi:10.3389/fphar.2016.00473
- Tang, C.; Yang, L.; Jiang, X.; Xu, C.; Wang, M.; Wang, Q.; Zhou, Z.; Xiang, Z.; Cui, H. Antibiotic drug tigecycline inhibited cell proliferation and induced autophagy in gastric cancer cells. Biochem. Biophys. Res. Commun. 2014, 446, 105-112. doi: 10.1016/j.bbrc.2014.02.043
- Hu, H.; Dong, Z.; Tan, P.; Zhang, Y.; Liu, L.; Yang, L.; Liu, Y.; Cui, H. Antibiotic drug tigecycline inhibits melanoma progression and metastasis in a p21CIP1/Waf1-dependent manner. Oncotarget 2016, 7, 3171-3185. doi: 10.18632/oncotarget.6419
Ad. 2. Following the Reviewer's suggestion, we conducted preliminary cytotoxicity analyses of tigecycline using human normal skin cells: melanocytes and fibroblasts. The study results and their discussion have been included in the revised version of the publication.
Results section:
“To assess control normal cell lines, preliminary analyses were conducted on human dermal fibroblasts and human epidermal melanocytes, darkly-pigmented. The test was performed after 48 h of incubation with tigecycline in the concentrations ranging from 1 to 400 µM. The obtained results (Figure 2) indicated that the drug, similar to melanoma cells, proportionally reduced the number of metabolically active cells with increasing concentration. However, the calculated EC50 values for the 48-hour treatment were higher compared to melanotic and amelanotic melanoma cells: 126.2 µM for HDF and 148.0 µM for HEMn-DP. It is also worth noting that only in normal darkly pigmented melanocytes, tigecycline at concentrations up to 25 µM did not exhibit cytotoxic effects.
Figure 2. The analysis of tigecycline cytotoxic potential on human dermal fibroblasts HDF and human epidermal melanocytes HEMn-DP. The cells were treated for 48 h. The results were expressed as a percentage of control. Mean values ± SD were presented. ** p < 0.005.”
Discussion section:
“The results of research showing the anti-melanoma potential of tigecycline were presented in this work. In the first stage, a preliminary evaluation of the cytotoxicity of the drug against amelanotic melanoma A375 and melanotic COLO 829 cells was conducted. The analyses showed that tigecycline reduced the number of metabolically active melanoma cells proportionally to the drug concentration and incubation time. The reduction was statistically significant already from a concentration of 1 µM. It was noticed that the observed decrease was greater in the melenotic type of cells. Taking into consideration the EC50 parameter, it was found that tigecycline exhibited stronger activity against melanoma cells compared to normal skin cells, especially melanocytes. The tested drug, at lower concentrations, up to 25 µM, did not caused a reduction in the number of metabolically active cells only in the case of melanocytes. In summary, it can be concluded that the ability of cells to synthesize melanin may have significant implications for the cytotoxicity of tigecycline, as well as the effectiveness and safety of the therapy with the analyzed drug. However, the relative resistance of melanocytes and the greater susceptibility of melanotic melanoma to tigecycline indicate that the role of melanins and melanogenesis is complex and ambiguous. In this regard, it is important to consider factors such as the type of melanin and its impact on redox homeostasis, cellular metabolism, inflammation development, as well as drug and toxin binding. The pleiotropic functions and properties of melanin contribute to its involvement in mutagenesis, immunosuppression, and tumor progression [35].”
- Slominski, R.M.; Sarna, T.; Płonka, P.M.; Raman, C.; Brożyna, A.A.; Slominski, A.T. Melanoma, Melanin, and Melanogenesis: The Yin and Yang Relationship. Front. Oncol. 2022, 12, 842496. doi:10.3389/fonc.2022.842496
Ad. 3. We would like to apologize to the Reviewer for any errors and inconsistencies. As suggested, we have made revisions to the manuscript according to the suggestions.
“The calculated EC50 values for COLO 829 cells were 91.5 µM, 21.0 µM and 19.2 µM for the respective treatment times: 24 h, 48 h, and 72 h.”
“It was observed that tigecycline-treated COLO 829 cells exhibited a tendency towards a rounded and spherical morphology, as well as independent growth without apparent intercellular interactions.”
Ad. 4. We would like to thank the Reviewer for the important comment regarding certain limitations in the use of tigecycline as an antibiotic drug. We have mentioned this issue in the paragraph added to the Discussion.
“The presented results indicate the therapeutic potential of tigecycline in the treatment of melanoma. The observed differences between melanoma types may be due to the involvement of melanins and melanogenesis in the cytotoxic effect of the drug. However, it should be taken into account that the tested cell lines were cultured in different media, which could have influenced the final outcome to some extent. Further studies confirming the effectiveness of tigecycline using other melanoma cell lines appear necessary. It is important to consider the genetic background, phenotypic characteristics of the cells, and a detailed analysis of the melanogenesis process. Conducting in vivo studies is also crucial, especially considering the significant interaction between neuroendocrine mechanisms regulating body homeostasis and melanoma cells [67]. It is also important to consider the antibiotic activity of tigecycline. This action can be significant for the effectiveness and safety of therapy in in vivo conditions. Furthermore, the spreading and increasing bacterial resistance to antibiotics, including tetracyclines, pose certain limitations for utilizing the non-antibiotic properties of tigecycline. Nevertheless, in justified cases, controlled treatment with tigecycline appears to be feasible.”
- Slominski, R.M.; Raman, C.; Chen, J.Y.; Slominski, A.T. How cancer hijacks the body's homeostasis through the neuroendocrine system. Trends Neurosci. 2023, 46, 263-275. doi:10.1016/j.tins.2023.01.003
Reviewer 4 Report
Rok et al. studied the anti-melanoma activity of tigecycline against amelanotic and melanotic melanoma cells. However, the study is not clear to prove the exact role of tigecycline. Moreover, previous proved that tigecycline as anti-melanoma property (Hu, H., Dong, Z., Tan, P., Zhang, Y., Liu, L., Yang, L., Liu, Y. and Cui, H., 2016. Antibiotic drug tigecycline inhibits melanoma progression and metastasis in a p21CIP1/Waf1-dependent manner. Oncotarget, 7(3), p.3171). Then what is the point of novelty in this study. Similarly, Hu et al. observed that 10 µM tigecycline is enough to inhibit the cell proliferation more than 50% but in this 400 µM showed more resistance. Results also more contradiction to previous report. Thus, the study should be deeper molecular insights to evaluate the manuscript for further.
Author Response
Response to the Reviewer #4
Dear Reviewer,
First and foremost, we would like to express our appreciation for reviewing our manuscript. We are grateful for bringing to our attention the publication by Hu et al. (doi: 10.18632/oncotarget.6419). The findings presented in that article are indeed intriguing, and we have decided to incorporate a new paragraph in the Introduction section to outline the key discoveries. We fully agree with the Reviewer that the disparity in the obtained EC50 values is surprising. However, it is important to note that Hu et al. assessed cell viability using the MTT assay, whereas in our study, we employed a different test, namely the WST-1 assay. Furthermore, the authors of the cited paper did not explicitly describe the timing of the treatment in relation to cell seeding. Although not clearly stated in the methodology, it could be inferred that the drug was added immediately after cell seeding, without a preceding incubation period ("Briefly, 2000 cells/well were plated onto 96-well plates then the MTT assay was carried out according to the manufacturer's protocol."). We would like to highlight that our preliminary cytotoxicity analyses also revealed a statistically significant effect of tigecycline at a concentration of 10 µM. However, none of the examined incubation times resulted in a decrease exceeding 50%. We suspect that certain discrepancies between our study and Hu et al.'s may arise from the variations in research methodologies.
Nevertheless, it is important to acknowledge that the pharmacodynamic activity of drugs is concentration-dependent. By employing lower concentrations of tigecycline, Hu et al. were able to demonstrate its anti-proliferative potential and investigate migration and invasion. In our study, we aimed to investigate whether tigecycline induced apoptosis or affected the process of autophagy in addition to inhibiting cell division. Drawing from our previous experience with tetracycline research, we selected higher concentrations for detailed analysis compared to Hu et al. Importantly, despite the relatively high concentration, tigecycline proved to be a much weaker inducer of apoptosis compared to minocycline and doxycycline. We believe that these observed differences among the drugs underscore the importance of studying individual tetracycline derivatives, which exhibit distinct pharmacological properties despite their similar structures. We consider the information obtained from our study to be a valuable guide for researchers planning to investigate tetracyclines.
It is crucial to emphasize that we fully recognize that the obtained results give rise to further questions, particularly regarding other molecular mechanisms underlying the action of tigecycline. The present study represents a preliminary analysis demonstrating the anticancer potential of tigecycline against melanotic and amelanotic melanoma cells. Our future investigations will focus on elucidating the roles of autophagy and melanogenesis in melanoma resistance to treatment.
We hope that these explanations are satisfactory and provide sufficient clarity for the Reviewer.
Round 2
Reviewer 1 Report
The authors adequately revised the manuscript
Minor editing is advised
Author Response
Response to the Reviewer #1
Dear Reviewer,
We would like to thank you for the proofreading and assessment of our article as well as for your kind opinion. We hope that the improved version of the paper is more reliable and will be satisfying for the Editors.
Reviewer 2 Report
The changes brought by the authors to the initial form of the article contribute to an increase in its scientific quality. At the same time, the changes support a better understanding of the results obtained by the authors.
minor changes
Author Response
Response to the Reviewer #2
Dear Reviewer,
First and foremost, we would like to express our appreciation for reviewing our manuscript and for a thorough and insightful assessment. We hope that the improved version of the paper is more reliable and will be satisfying for the Editors.
Reviewer 4 Report
The author justified the research work with previously published data Hu et al. 2016. However, there are few minor queries need to be clarified.
1. In the cytotoxicity study, after 48 h metabolic active cells were decreased in COLO829 and A375 cells. whereas, in case of viability study the percentage of cell death (fig.3 and fig.4) quite less what could be the reason. Please clarify
2. Please check abbreviations once because it was repeated in few places.
3. Cite the sub figures (Fig. 1a/fig.1b) in the manuscript
4. Check once again chemicals brand name, company, country procured details
Author Response
Response to the Reviewer #4
Dear Reviewer,
We are grateful for your valuable comments and suggestions. We apologize for the inaccuracies that appeared in the manuscript. In response to the review, we have removed the repetitive abbreviations and made corrections to the Materials and Methods section. Some reagents did not have a specified supplier, and these details have been provided. As suggested, we have cited the subfigures in the text of the publication.
Regarding the first comment, we would like to emphasize that WST-1 is a screening test used for an initial general assessment of drug-induced toxicity. However, considering the principle of the assay, it should be noted that the differences between the control and experimental groups may arise from both an antiproliferative effect (cytostatic action) and a cytotoxic effect leading to cell death. Ultimately, we observe only differences in the number of metabolically active cells. Additionally, WST-1 relies on the activity of mitochondrial dehydrogenases, which may vary in sensitivity to different factors. In conclusion, similar to other tests, WST-1 also has certain limitations. Therefore, in our opinion, additional analyses are always necessary to indicate the cytotoxicity mechanism in a more detailed manner. Nonetheless, tests such as WST-1 are highly useful in the initial stage of research, allowing for the calculation of EC50 values and comparisons between different drugs, concentrations, and cell lines. In the revised version of the manuscript, we have included an explanation regarding WST-1.
„It is worth noting that the results of the WST-1 test are an outcome of both the antiproliferative and cytotoxic effect, leading to cell death. Therefore, additional analyzes indicating the mechanism of action of the drug more precisely are necessary. Considering the results, next experiments were conducted using selected drug concentrations: 200 µM and 400 µM for the A375 cell line and 100 µM and 200 µM for COLO 829 melanoma cells.”
Thank you once again for your valuable remarks.